# Determination of Shear Capacity for Load Rating of Concrete Bridges to AS 5100.7-2017

**Koon Wan Wong [1],[†] and Vanissorn Vimonsatit [2],[*],[†]**

1    Retired Bridge Structural Engineer, Perth, WA 6147, Australia
2    Civil Engineering at the School of Engineering, Faculty of Science and Engineering, Macquarie University, Sydney, NSW 2109, Australia
*    Correspondence: sorn.vimonsatit@mq.edu.au
†    These authors contributed equally to this work.

**Abstract:** According to Modified Compression Field Theory (MCFT), the ultimate shear capacity of a reinforced concrete section depends on load effects (shear, moment, torsion, and axial force) caused by factored design loads. In many design standards, including Australian AS 5100.7, MCFT has been incorporated for bridge assessment, which requires a load rating to be carried out according to the loading of the nominated rating vehicle as prescribed in the standard. Recently, some approaches have been proposed for bridge load rating that have suggested using an iterative-search procedure to determine the shear capacity by proportionally increasing the load effects until the shear capacity and shear are equal. This paper describes several adverse effects of using the proportional load, which is not consistent with the characteristic of the vehicle loading, to determine the shear capacity for load rating. Numerical examples of two bridge beams, one simply supported and the other continuous, are presented to demonstrate that the characteristic of the load effects caused by a moving vehicle is not representable by proportional load effects. Furthermore, the current practice in the bridge load rating does not load rate the longitudinal steel capacity in resisting the axial force induced by the load effects of the rating vehicle. This paper presents a new approach to the load rating that separately accounts for the load effect for axial failure mode of the longitudinal steel. Finally, it is pointed out that locating the critical section where the rating factor is minimum is tedious but can be automated by integrating load rating into the analysis of load effects.

**Keywords:** concrete bridges; shear capacity; load rating; modified compression field theory

## 1. Introduction

Major design standards such as AASHTO Manual for Bridge Evaluation [1] and the Canadian Highway Bridge Design Code [2] have provisions for bridge strength assessments and load rating using modified compression field theory (MCFT). In the Australian set of standards for bridges AS 5100 series, the shear provisions were revised in 2017 to use MCFT equations. MCFT gives a more accurate determination of ultimate shear capacity than the approaches used in older standards. A major change is that shear capacity is now dependent on several load effects including shear, moment, torsion and axial force. AS 5100.5 [3] is used for design and AS 5100.7 [4] for bridge assessment. For load rating of concrete structures, AS 5100.7 refers to AS 5100.5 for the calculation of capacity.

The current practice in Australia for load-rating is to determine shear capacity directly using the longitudinal strain at mid-depth of the section from factored load effects (e.g., moment, shear, torsion, axial force). The provision to limit steel stresses (by limiting forces) in longitudinal reinforcement, as stipulated in Clauses 8.2.7 and 8.2.8 AS 5100.5, is suitable for design. However, AS 5100.7 does not provide information to load rate this effect. Caprani and Melhem [5] noted that anecdotal observations of consultants' reports showed that the load effect for failure mode of yielding longitudinal steel was not load rated. In the report

by Holt et al. [6], the state agency identified as "State B" noted that the Manual for Bridge Evaluation [1] did not give any guidance on the actions to take when the check on tensile capacity of the longitudinal steel on the flexural tension side of a member failed, and that they could not arrive at a consensus on the actions to take when that occurred in load rating. Not including the rating of this effect results in inadequate information available for strengthening bridges. It should be noted that even if a check on the adequacy of the longitudinal steel provided is carried out at the load effect level of the factored design shear, $V^*$, similar to the design check, it only provides an indication of adequacy but not the extent since the rating factor for the load effect of this failure mode is not determined.

Caprani and Melhem [5] suggested that the calculation of ultimate shear capacity $V_u$ directly as stipulated in AS 5100.7 without the use of iterations was not suitable and they recommended that an additional clause be included in this standard to use their proposed approach which required iterations. The reason provided was that the assessment discussed by Collins and Mitchell [7] and others (which were not referenced to sources in the published article) uses iterations. It should be noted however that Collins and Mitchell predicted the failure shear of a beam with a monotonically increasing test load which required the use of an iterative solution search procedure to predict the failure shear $V_{um} = V_n$. The letter "$m$" is used to indicate that the strength was calculated using mean material properties and the letter "$n$" is used to indicate nominal (unfactored) load effects. This iterative procedure has to be used as the level of load effects to cause failure is not known beforehand. In contrast, the calculation of ultimate shear capacity for load rating to AS 5100.7 does not require the use of iterations because the assumed live load is one or more of the same rating vehicle moving on a bridge where the load effects can be determined at any stage of loading.

In testing, iterate to predict the mean strength $V_{um}$ equals nominal shear represents the physical condition of increasing load to cause shear failure. The general expression $V^* \leq \phi V_u$ in AS 5100.5 represents a condition where the shear capacity is adequate to the requirement of AS 5100.7. The factor $\phi$ in this expression is the shear capacity reduction factor for design using linear elastic analysis. In contrast, the proportionally increased shear to determine $V_u$ to be the capacity equal to $V^*$ does not represent any key condition in the process of load rating. This is because the assumed loading is not as prescribed in AS 5100.7, and therefore the determination of capacity does not satisfy a key feature of MCFT of shear strength being a function of the applied load.

Furthermore, the load effects of two failure modes were considered in the same rating [5], which are the section shear and the yielding of the longitudinal reinforcement from the force caused by several load effects. The minimum shear capacity from the assumed proportional loading was selected to use for load rating. This is not advisable in practice because the load effects influencing these modes should be rated separately to give a rating factor for each mode.

Zheng et al. [8] stated that bridge load rating involves performing a series of calculations synonymous with those of bridge design in order to determine whether a bridge is safe for public traffic loads. This view is in line with the general understanding of the philosophy of load rating. Contrary to this, the use of an iterative solution-search procedure to determine capacity for load rating is markedly different to the method for the determination of capacity for design.

Since the iterative approach uses an assumed characteristic of loading not that stipulated in AS 5100.7, it is important to understand the adverse effect of using the assumed loading instead of the prescribed loading of AS 5100.7. This is the aim of the present study.

In addition, this paper addresses shortcomings in the load rating where the load effect for the failure mode of yielding of longitudinal steel caused by several load effects was not load rated. In Section 4, an approach is recommended which is suitable for considering the forces in the longitudinal steel reinforcement without the use of iterations in the load rating thus enabling this failure mode to be included.

## 2. Background

### 2.1. Load Rating

Strength assessment of a concrete bridge to AS 5100.7 requires a load rating factor described in {Section 14.1 of AS 5100.7 to be determined for all strength checks (e.g., moment, shear) for a nominated rating vehicle (see Section 14, AS 5100.7). For each loading effect, the factor is the lowest at all potential critical sections including those listed in Clause 10.7 of AS 5100.7. The relevant road authority managing the bridge often specifies several rating vehicles and provides guideline on load rating, which is to be carried out on both new and existing bridges. Rating factor is determined using the ultimate shear capacity as described in Section 10.6.2 of AS 5100.7 for concrete members. The determination of shear capacity based on MCFT requires the use of the longitudinal strain in the concrete at mid-depth from the combined effects of the live (vehicle) load and permanent load acting on the bridge.

In this paper, the symbols $M^*$, $V^*$, and other load-effect terms with an asterisk in the superscript represent factored values, not unfactored in AS 5100.7. This is to be consistent with AS 5100.5 (and AS 3600 [9]) in which the superscript asterisk represents factored values. For the design of a new bridge to AS 5100.5, these effects are generally referred as design actions caused by design (factored) loads. In a bridge assessment, the load effects do not necessary have the same values as the design actions considered in the bridge design for the same vehicle. Live load factors may be reduced in situations where the speed of the vehicle is specified. They may also be modified if approved by the relevant authority where appropriate significant measurement is undertaken. Material properties for rating may be different from those used for the design of a bridge owing to deterioration since built.

Owing to the novelty of the use of MCFT-based provisions for bridge load rating for shear, literature on the assessment of MCFT-based shear capacity for load rating is still limited. A report by Holt et al. [6] describes the challenges and difficulties in evaluating shear capacity for load rating of concrete bridges using the Manual for Bridge Evaluation [1] which uses the shear provisions in the Standard Specification for Highway Bridges [10]. They highlighted the importance of ensuring that the shear resistance is consistent with the longitudinal strain from the applied load. However, the practice of using maximum values obtained from envelopes of factored load effects to calculate the longitudinal strain (for calculating shear capacity) of a section while is suitable (and conservative) for design, is not for load rating. For accurate load rating, a consistent set of coexisting load effects should be used.

Currently, to the authors' knowledge, major road design standards such as AASHTO design specifications do not have provisions to use an iterative approach to determine the shear capacity for load rating. LRFD (Load and Resistance Factor Design) in the US standards is similar to the Limit States Approach in Australian standards, and both AASHTO Standard Specifications for Highway Bridges and AS 5100.7 use shear formulations based on MCFT.

### 2.2. Consideration of Shear Action and Shear Capacity

2.2.1. Effect of Proportional Loading Approach on the Load Rating of New Bridges

The load rating of new bridges using the design vehicle as the rating vehicle is often required by road authorities. For example, Section 2.4.1 of the Bridge Branch Design Information Manual of Main Roads Western Australia [11] requires all new structures to be rated for the vehicles shown in document DIS 3912/02-4, which is the Structures Engineering Design Manual of Main Roads Western Australia [12]. Section 4 of the Structures Engineering Design Manual provided a list of design vehicles for new bridges. Also, AS 5100.7 Section 11.3.1 stated that a design vehicle might be included as one of the nominated rating vehicles.

The proportional loading approach [5] assumed a loading, which shear force starts from zero, is monotonically increasing, and is proportional to the other load effects. They assumed the absolute ratios, termed in the paper as influence coefficients, between shear

force and the other action effects (bending moment, torsion and axial force) remain constant. These coefficients were determined from the load effects caused for the most critical load case. An iterative numerical search procedure was used to determine the shear capacity equal to the shear force.

This approach results in the shear capacity determined for load rating of a new bridge being different from that calculated for design for the same vehicle. For new bridges, the ultimate shear capacity determined for bridge assessment to AS 5100.7 is expected to be the same as the shear capacity determined for design to AS 5100.5, and this expectation is not met when using the proportional loading approach.

### 2.2.2. Shear Load Rating and Load Effects

In structures with several possible failure modes, it is a usual practice to provide load rating information for all modes in addition to identifying which mode is more limiting and to what extent. It is particularly important to provide this useful information in order to manage the strengthening of existing bridges.

The failure caused by section shear and the failure caused by yielding of the longitudinal reinforcement are two separate modes. The axial force in the longitudinal steel was included [5] by calculating the level of shear for the monotonic loading to cause yielding of the steel. Since the lower of two shear limits was assumed to be the shear capacity for load rating, the load rating factor of the less critical effect was not reported and the resulting rating factor does not provide information as to which of the two influencing load effects (section shear and axial force) is more critical.

### 2.2.3. Ultimate Shear Capacity for Load Rating

The ultimate section shear capacity for load rating to AS 5100.7 is determined using characteristic material properties. In contrast, the prediction of shear strength at failure of a test structure uses mean values instead of characteristic values.

The ultimate shear capacity $V_u$ of several reinforced and prestressed concrete beam sections were determined [5] using an iterative solution procedure to determine $V_u = V^*$ for a proposed incremental proportional load effect $V^*$ which was assumed to start from $V^* = 0$.

Another iterative approach was recommended by Holt et al. [13] to determine the section shear capacity by assuming that only the factored live load component from the rating vehicle was monotonically increasing. The load effects of three failure modes, namely section shear, axial force of the longitudinal reinforcement, and interface shear, in determining the shear capacity were considered instead of two, i.e., section shear and axial force [5]. The use of this approach also introduces inaccuracies to the load rating because the load effects in this case are still not consistent with the loading of the rating vehicle. Several papers which described the use of iterations for strength determination were cited to support their use of iteration for load rating. It should again be pointed out that the iterative approach is not about the determination of shear capacity for load rating, but for the prediction of the shear at failure of test structures subjected to increasing test loads.

## 3. Present Approach to Shear Load Rating

In the present work, the rating factor for section shear is determined in accordance with AS 5100.7. The calculation is automated by considering the capacity and load rating factors for the potential critical section for the entire loading history of the vehicular movement. The load rating for each moving step is determined immediately after the structural analysis using the load effects in that step.

Before discussing the effect on shear capacity of the present approach in comparison with that of the proportional loading approach for load rating, an understanding of the characteristic of the load effects from a moving vehicle on a bridge is important. Engineers carrying out load rating for shear in the past often used shear envelopes to identify

the critical section. This is because the shear capacity in old design standards did not depend on several co-existing load effects. To demonstrate this point, line beam analysis was carried out for a simply supported bridge and for a two-span continuous bridge to show the action effects from a moving vehicle. The analysis and results are presented in Sections 3.1 and 3.2 respectively.

*3.1. Analysis*

The analysis in this study uses a line beam model for two structures subjected only to self-weight and a moving rating vehicle, one simply supported and the other continuous with a single inner support. Only the load effects of shear and moment are considered. This simplified model for both the structures and the live loading are acceptable for the purpose of this study to provide an indication of the characteristic of the load effects. This simplification allows the inter-relationship of three variables, shear, bending moment and total longitudinal strain in the concrete to be shown graphically.

A computer program written in Python [14] was developed to carry out the structural analysis and to calculate the ultimate shear capacity directly from the load effects as loading progresses. The vehicle was moved along the beam in steps of 0.1 m, and for each step, the locations of the axles on the beam were determined, and the Hardy-Cross moment distribution method [15] was used to calculate the load effects at the potential critical section for the two-span continuous beam. The capacity terms $V_u$ and $\phi V_u$ were calculated immediately after each determination of action effects in a step. Also the rating factor for force in the longitudinal reinforcement using the approach described in Section 4 and the rating factor for section shear as described in AS 5100.7 were also calculated. Matplotlib [16] was the python module used to produce several graphical displays on screen to enable visualisation of the result at the end of the loading. The software Octave [17] was used to produce the quality plots presented in this paper using the results obtained from the analysis.

*3.2. Load Effects from a Moving Vehicle*

The load effects and their corresponding ultimate shear capacity have been determined in accordance with AS 5100.7 for a single 8-m span, simply-supported bridge beam as shown in Figure 1 with the factored axle loads acting on the beam from the rating vehicle as shown in Figure 2. The axles loads are for Group 1 Vehicle 3 (G1V3), with an axle group configuration of 1-2-2-4, which is one of the load rating vehicles of the road authority in Western Australia [11]. The nominal (unfactored) axle load of the first axle is 6 tonnes and each remaining axles is 9 tonnes. For simplicity, only self weight is included in the permanent load in the analysis to show the general characteristic of the load effects due to the moving vehicle on the bridge.

In practice, load rating to AS 5100.7 requires full compliance to this standard. A more complex bridge model (e.g., using a grillage or finite elements) is required to give accurate live load distribution. All permanent loads and accompanied vehicles (if required) have to be included. Furthermore, requirements for load rating issued by the relevant road authority must be followed. In the present study, the load from the 9 tonne axle acting on an inner beam was calculated using a live load factor of 2.0, a dynamic load allowance factor of 0.4, and a live load distribution of 0.5 using the Lever Rule. Thus, the concentrated factored load on the beam from each axle of 9 tonnes is estimated to be 124 kN.

The concrete bridge is assumed to be of monolithic construction with an overall depth of 450 mm, a slab thickness of 150 mm with 300 mm webs spaced 850 mm apart (i.e., $D = 450$ mm, $d_f = 150$ mm, $b_f = 850$ mm, $b_v = 300$ mm). The dimensions of the cross-section are shown in Figure 3. The main reinforcement is 3N24 ($A_{st} = 1350$ mm$^2$), with 2 legged ligatures (stirrups) of size N10 with a uniform spacing of 350 mm ($A_{sv} = 160$ mm$^2$, $s = 350$ mm). Material properties are $f'_c = 40$ MPa, $f_{sy} = 500$ MPa and $f_{sy.f} = 500$ MPa. Effective depth $d$ is assumed to be 400 mm. Factored self weight is 6.51 kN/m, calculated using a load factor of 1.2, cross section area of 0.217 m$^2$ and the unit weight of reinforced

concrete is 25 kN/m$^3$. The amount of ligatures (stirrups) is chosen to be light to show the load effects for a shear deficient structure. The section was assumed to behave as a T-beam in bending for the calculation of the lever arm $z$ and a rectangular beam for shear.

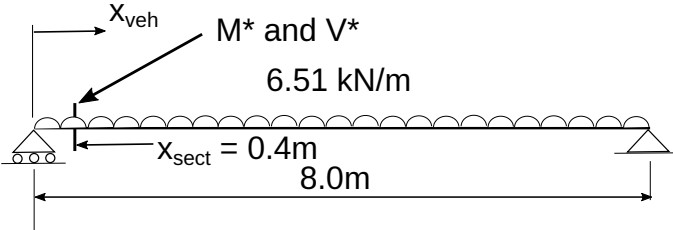

**Figure 1.** Simply-supported bridge beam with factored self-weight.

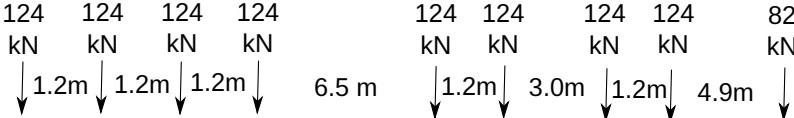

**Figure 2.** Factored axle loads acting on a beam from the 1-2-2-4 vehicle.

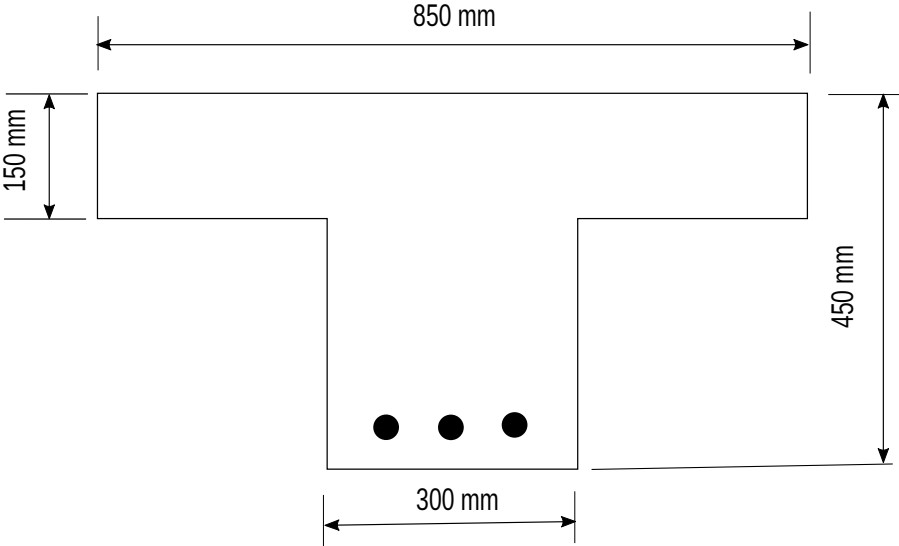

**Figure 3.** Cross-section of bridge beam.

Consider the rating vehicle moving from the left end of a simply supported bridge beam shown in Figure 1, the factored load effects $M^*$ and $V^*$ and the reduced ultimate shear capacity $\phi V_u$ are determined in accordance with AS 5100.7 at a critical section 0.4 m from the left support. The capacity reduction factor $\phi$ for shear from AS 5100.5 is 0.7. The analysis was carried out for the vehicle movement over its entire range of the movement from the vehicle entering the bridge ($x_{veh}$ = 0) to leaving it ($x_{veh}$ = 28.4 m). The notation $x_{veh}$ is the distance of the front axle of the vehicle from the left support. In the analysis, an incremental movement step is 0.1 m. When an axle is passing the critical section ($x_{sect}$ = 0.4 m), a small numerical distance of 0.01 m, before and after the section, was considered to enable shear discontinuities to be obtained in the analysis.

Figure 4 shows the shear $V^*$ of the section due to the moving load effect of the vehicle for $x_{veh}$ = 0.0 to 20.8 m, the last position is when the last axle just crosses the section, which is plotted against the longitudinal strain in concrete at the mid-depth of section, $\epsilon_x$. This figure shows that shear first decreases when an axle moves along the shear span and abruptly increases when it crosses the section. The reduced ultimate shear capacity $\phi V_u$ is also plotted in the same figure. The analysis shows that the capacity is exceeded (i.e.,

$V^* > \phi V_u$ ) when the third axle crosses the section. The lowest $\phi V_u$ is 142 kN, which occurs when $V^*$ is 382 kN and $M^*$ is 153 kNm.

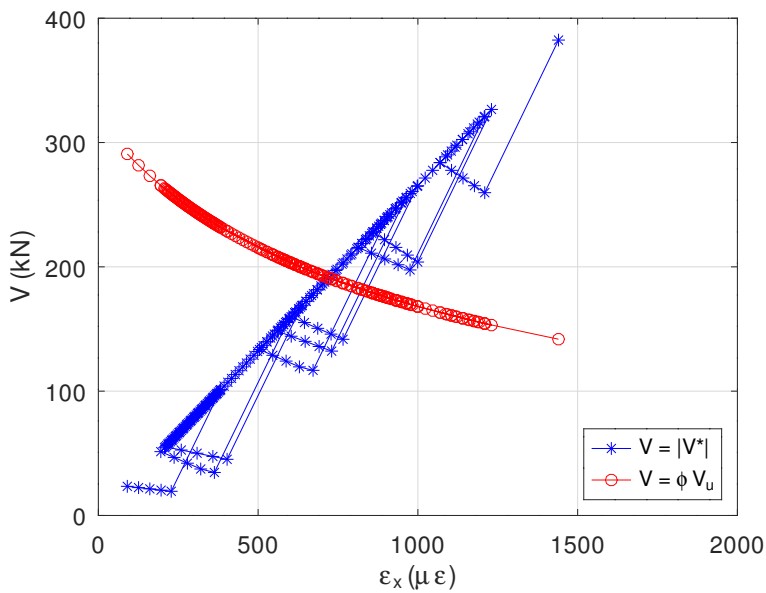

**Figure 4.** $V^*$ versus $\epsilon_x$ for $x_{veh} = 0.0$ to 20.8 m.

A plot of $V^*$ against $M^*$ is shown in Figure 5. This plot shows the discontinuities in shear when an axle crosses the section owing to the assumption of modelling the axles as concentrated axle loads. This assumption is commonly used with a line beam model. In reality, each wheel occupies a finite area on the bridge deck. The shear increases abruptly when an axle crosses the section. Thus there is no clear intersection between the shear $V^*$ with its corresponding reduced capacity $\phi V_u$ line. Even if the axles are modelled using a uniform distributed load, the increase in shear will still be abrupt since the length of the wheel patch in the direction of travel is only 0.2 m.

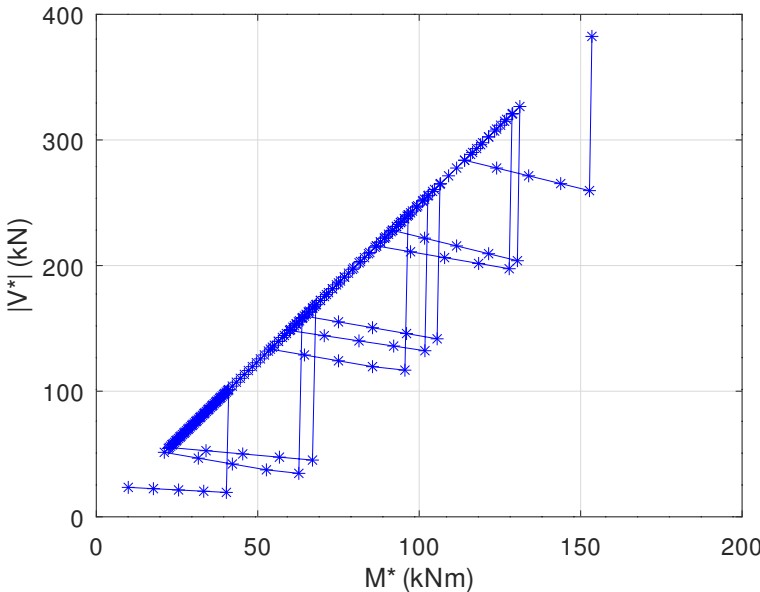

**Figure 5.** $V^*$ versus $M^*$ for $x_{veh} = 0.0$ to 20.8 m in steps of 0.1 m.

The load effects from a moving vehicle on a bridge are quite complicated. This can be seen from the 3D plot shown in Figure 6. The movement plotted is for $x_{veh}$ from 16.8 m to 20.8 m, the movement of the last axle group on the shear span to end just past the section.

These load effects are quite different from those caused by a monotonically increasing load applied to cause structural failure in shear in a load-testing situation.

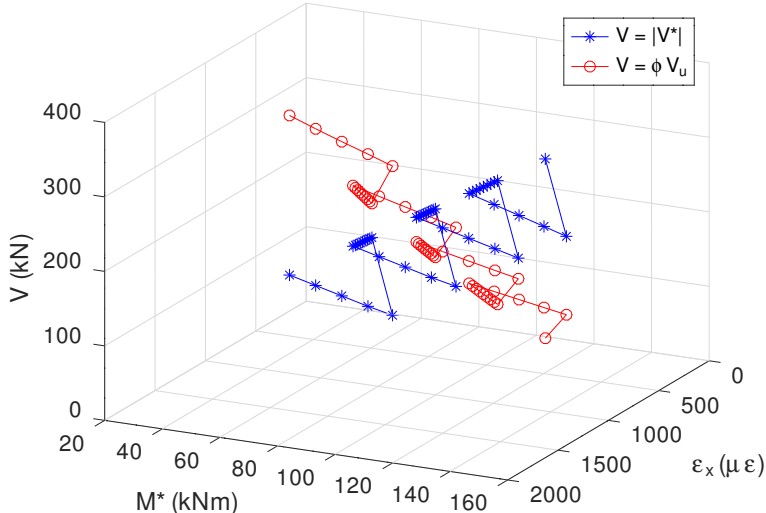

**Figure 6.** 3D plot for $x_{veh}$ = 16.8 to 20.8 m in steps of 0.1 m.

Owing to the requirement for equilibrium, the values of ratio $|M^*|/|V^*|$ immediately after shear discontinuities are close to 0.4 m, the length of the shear span, since no axles fall within the shear span and the contribution to these load effects from the self-weight within the shear span is relatively small. The values of the ratio are not constant for the two-span continuous beam shown in Figure 7 for a nominated vehicle with factored axle loads shown in Figure 8 . The axle loads and the spacings are the same as the nominated G1V1 rating vehicle of MRWA [11] except one of the inter-axle spacings of 7.25 m was changed to 7.2 m to ensure every axle gets to land exactly on the critical section during the analysis. The potential critical section is 0.4 m to the right of the internal support. The dimensions of the section, the amount of reinforcement on the tension side of the section, and the amount of ligatures (stirrups) steel reinforcement are assumed to be the same as those for the simply supported beam described earlier in this paper. Owing to negative bending moment, the section is assumed rectangular for both bending (to determine $z$) and shear. The relationship between $|V^*|$ and $|M^*|$ for $x_{veh}$ = 10.0 to 34.5 m is shown in Figure 9, for the vehicle starting at the inner support to travel on span 2 until it leaves the bridge. The ratios of $|M^*|$ to $|V^*|$ at the end of shear discontinuities do not have the same value.

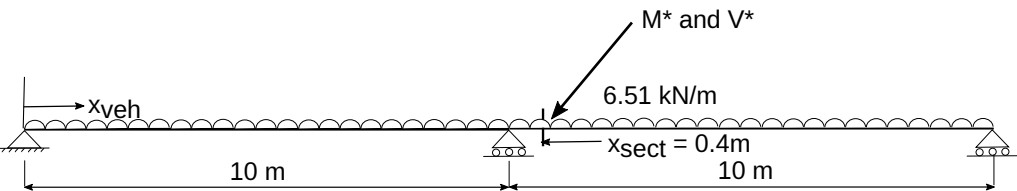

**Figure 7.** Two-span bridge beam with factored self-weight.

124 kN  124 kN        124 kN  124 kN        82 kN

|1.2m|        7.2m        |1.2m|  4.9m

**Figure 8.** Factored axle loads acting on a beam from the 1-2-2 vehicle.

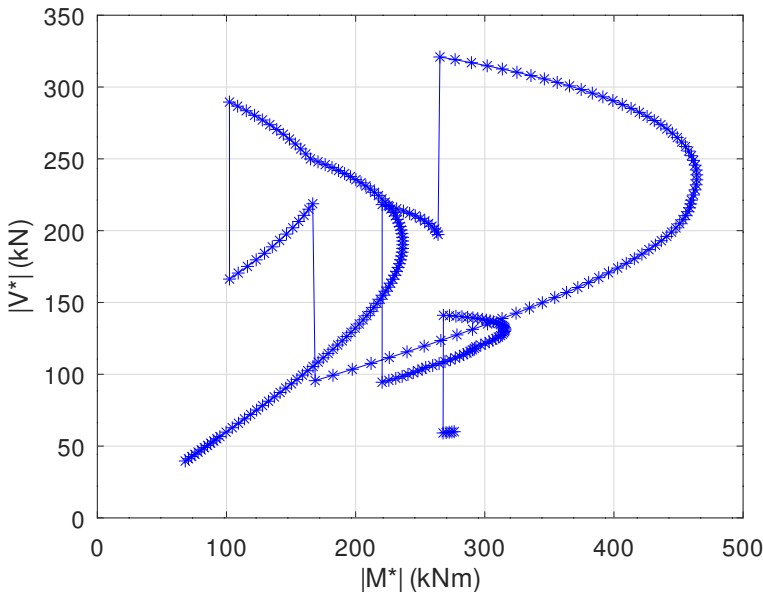

**Figure 9.** $V^*$ versus $M^*$ for $x_{veh}$ = 10.0 to 34.5 m in steps of 0.1 m.

Figure 10 shows the shear $V^*$ of the section which is plotted against the longitudinal strain in concrete at the mid-depth of section, $\epsilon_x$. The reduced ultimate shear capacity $\phi V_u$ is also plotted in the same figure. The analysis shows that the shear capacity is exceeded (i.e., $V^* > \phi V_u$) when the first axle crosses the section. $|M^*|$ is 286 kNm, $|V^*|$ is 141 kN and $\phi V_u$ is 126 kN.

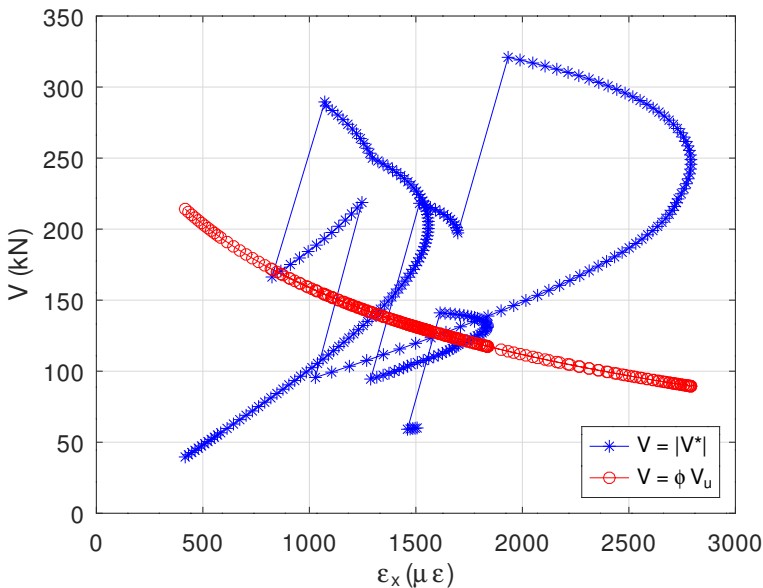

**Figure 10.** $V^*$ versus $\epsilon_x$ for $x_{veh}$ = 10.0 to 34.5 m in steps of 0.1 m.

Load rating factors for shear were calculated using the load effects directly without the use of iterations at every step of movement, and they are shown in the plot of load rating factor versus $x_{veh}$ in Figure 11 for the vehicle-bridge system for the continuous beam. Only factors between zero and 0.8 are shown in the plot. The lowest rating factor is 0.23 when $x_{veh}$ is 18.0 m. It did not occur at the end of the shear discontinuity when the third axle crossed the critical section at $x_{veh}$ = 16.5 m. It occurs at maximum absolute moment with decreasing shear after the axle moves away from the section.

In AS 5100.7, both equations for the calculation of shear capacity are not applicable for $\epsilon_x > 3 \times 10^{-3}$. Figure 10 shows that the largest absolute value of the longitudinal

strain is only slightly below this limiting value. For design, when the calculated strain exceeds this value, reinforcing steel can be increased or a larger section can be used, which is not possible in load rating. When this occurs in load rating, the relevant road authority should provide guidance as to the value to assign to the rating. In bridge management information systems of road authorities, usually a rating value has to be assigned to a bridge for nominated rating vehicles, and one way to take this into consideration is to specify the load rating factor for shear in this case to be zero to show that the vehicle-bridge system cannot be load rated using the equations of AS 5100.7.

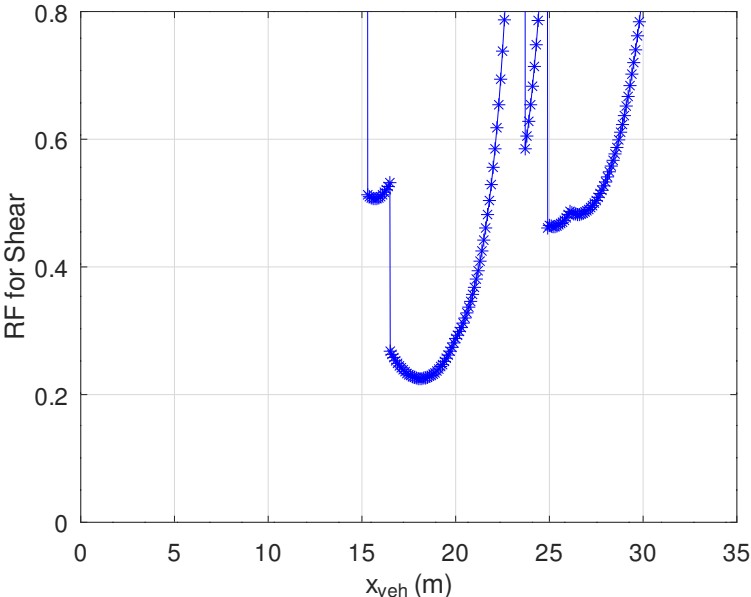

**Figure 11.** Rating Factor for shear versus $x_{veh}$ for the two-span beam.

## 4. Recommendation for Load Rating to Consider the Failure Mode Due to Yielding for Axial Forces in Longitudinal Steel

As described in Section 1, load rating to consider the failure mode due to the yielding of the longitudinal reinforcement from the force induced by several load effects has not been properly carried out. Load rating of this effect through shear without the use of a fictitious incremental loading is difficult as shear is one of the several independent variables in the non-linear MCFT equations. Since this axial force is a function of several load effects, including shear, moment, axial force and torsion ($V^*$, $M^*$, $N^*$, $T^*$), rating its effect through shear is not logical as one or more of the other load effects might be more dominant. While the use of the incremental loading enables this effect to be included, using the limiting shear capacity from a load effect which is not consistent with that caused by the actual loading introduces inaccuracies to the load rating.

A new approach is to consider the total force in the longitudinal reinforcement as a load effect, similar to the other effects (e.g., $M^*$, $V^*$), that has to be load rated using the yielding force as the capacity. This load effect is different from the other load effects in that it is a function of several other load effects and the other terms as given in Equation (3).

However, the current equations which use additional force in Clauses 8.2.7 and 8.2.8 in AS 5100.5 are not suitable for the load rating of this effect. They should be changed to using the total force, as in AS 3600 [9]. Following the notations used in AS 3600, the longitudinal steel for the flexural tension side is to be provided to resist the total tension force $T_{td}$ as given by Equation 8.2.8.2(1) in AS 3600, reproduced as Equation (1). Similarly, Capacity $T_{td.capacity}$ is given by Equation (2) which is a rearrangement of Equation 8.2.8.2(2) in AS 3600, and replacing the inequality sign with an equal sign. Also, the term $f_{st}$ is replaced by $f_{sy}$ assuming that the steel is fully anchored. The terms $A_{pt}$ and $\sigma_{pu}$ are related to prestressed steel reinforcement. The value of the capacity reduction factor $\phi$ in this equation is 0.7 from AS 5100.7. The text "capacity" is added to the subscript to

denote the capacity. Using these equations, the load rating factor for the axial force in the longitudinal steel in the flexural tension side can be calculated. The term $\Delta F_{td}$ is the additional longitudinal tension force caused by shear given by Equation 8.2.7(2) for shear without torsion in AS 5100.5 and reproduced as Equation (3). The notations in this equation are defined in AS 5100.5, in which $\theta_v$ is the angle between the axis of the compression strut and the longitudinal axis of the member. For reinforced concrete section, the term $\gamma_p P_v$ for prestressed concrete member is not applicable, thus Equation (3) is reduced to Equation (4). Similar equations can be written for the steel in the flexural compression side of the concrete section.

$$T_{td} = \frac{M^*}{z} + \frac{N^*}{2} + \Delta F_{td} \tag{1}$$

$$T_{td.capacity} = \phi\left(A_{st} f_{sy} + A_{pt} \sigma_{pu}\right) \tag{2}$$

$$\Delta F_{td} = \cot(\theta_v)\left(|V^*| - 0.5\phi V_{us} - \gamma_p P_v\right) \tag{3}$$

$$\Delta F_{td} = \cot(\theta_v)\left(|V^*| - 0.5\phi V_{us}\right) \tag{4}$$

As an example, the calculation to determine the rating factor of the section at $x_{sect}$ = 0.4 m for force in the longitudinal steel on the flexural tension side of the beam for the problem shown in Figure 1 is presented in Appendix A. The section is assumed to behave as a T-beam in bending and a rectangular beam in shear.

The rating factors for force in the longitudinal reinforcement on the flexural tension side of the critical section for the simply-supported bridge beam for all movement steps are shown as a plot of the rating factor versus $x_{veh}$ in Figure 12. Only load factors below 2.0 are shown in the diagram. From the analysis, the minimum value is 0.563 which occurs when $x_{veh}$ is 20.81 m. This value is close to the value of 0.57 calculated in Appendix A.

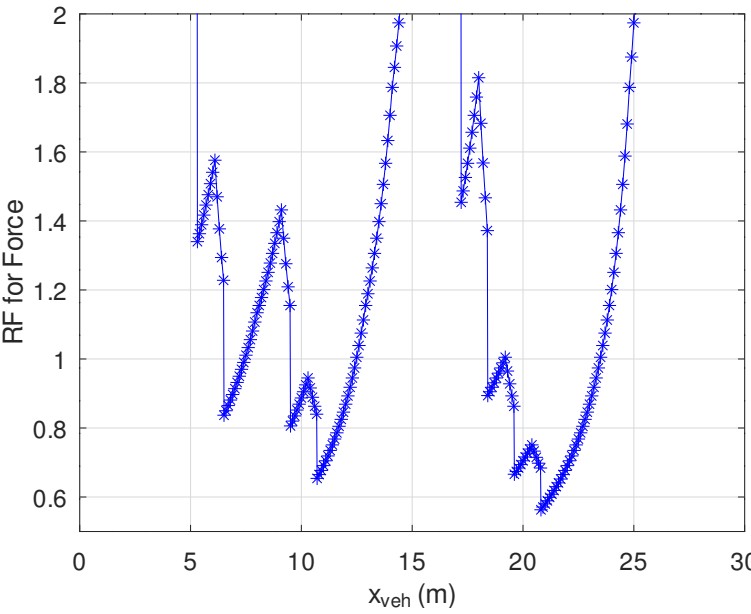

**Figure 12.** Rating Factor for force versus $x_{veh}$ for the simply-supported beam.

The load rating factor for force in longitudinal reinforcement on the flexural tension side of the critical section for the two-span bridge beam for all movement steps are shown as a plot of rating factor versus $x_{veh}$ in Figure 13. Only factors below 0.6 are shown in the diagram. From the analysis, the minimum value is 0.225 which occurs when $x_{veh}$ is 18.1 m.

Load rating the load effects of shear and longitudinal forces separately provides useful information which allows road authorities to determine whether the deficiency of a bridge is mainly due to insufficient longitudinal reinforcement or insufficient shear reinforcement. The issue with using the load rating equations in AS 5100. 7 is that there is no information provided on how to load rate the forces in the longitudinal steels which have equations with non-linear terms. The standard for bridge assessment AS 5100. 7 should be amended to address this deficiency.

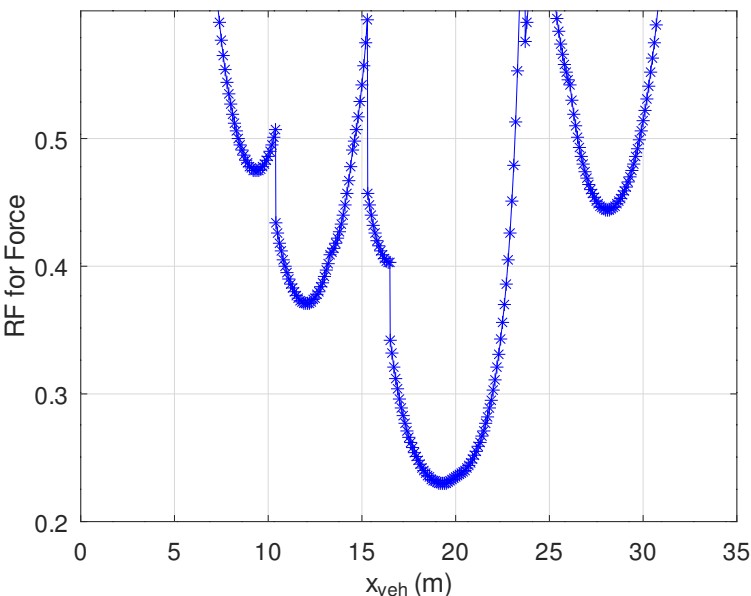

**Figure 13.** Rating Factor for force versus $x_{veh}$ for the two-span beam.

Longitudinal shear in composite beams and monolithic beams with flanges can similarly be load rated using the interface shear flow (in kN/m) at the interface. The factored shear stress from Equation 8.4.2 of AS 5100.7 is multiplied by the width of shear plane, $b_f$, to give this value. Similarly the shear stress capacity $\tau_u$ determined using Equation 8.4.3 of AS 5100.7 is multiplied by this width and the capacity reduction factor $\phi$ to give the reduced shear flow capacity (in kN/m).

## 5. Automating Load Rating of Shear

Identifying the critical section in the flexural region due to moving vehicle load, where the lowest load rating factor for shear occurs, is not a straightforward task because the ultimate shear capacity calculated using MCFT-based equations depends on several load effects. The critical section might not occur at the section with the maximum shear identified from the shear envelope of the moving load. While Holt et al. [6] suggested considering also both maximum positive and negative moment and the coexisting shear for indeterminate beams, the critical section can still be missed. It is possible for the lowest load rating factor to occur at a section with a combination of moment and shear that neither absolute shear nor moment is maximum. This task of manually selecting sections to load rate is more risky in cases where one or more of the co-existing action effects is discontinuous, longitudinal reinforcement is not the same (owing to bar curtailment), the shear reinforcement is not uniform, and either the width or depth or both vary along the beam. Automating the process of shear load rating will reduce, if not totally eliminate, this risk.

Furthermore, in addition to reducing human errors, automating the process improves efficiency, especially where numerous rating vehicles are nominated by road authorities. For example, Main Roads Western Australia [11] requires the inclusion of four Group 1 and six Group 2 rating vehicles for load rating of new and existing concrete bridges.

Integrating the load rating process into structural analysis enables load rating factors for shear at numerous sections to be calculated for the rating vehicle moving on the bridge.

It is also possible to use an iterative solution-search procedure to locate accurately the section with the lowest load factor. Such procedures (e.g., the "search and halving" procedure described in the report by Ahmad and Warner [18]) have been used in previous studies to solve nonlinear problems. They work well for problems without discontinuities. As the locations where discontinuities described above occur are known or can be determined during the structural analysis, the beam can be subdivided into segments between discontinuities and the lowest load rating factor is first determined for each segment. The procedure must be capable of finding the section with the minimum loading factor for shear within each segment. The lowest of these values is then selected to be the lowest rating factor for the beam.

## 6. Adverse Effects of Using Iteration to Determine the Shear Capacity for Vehicle Load Rating

As described earlier in this paper, using an iterative approach and the assumed loading not consistent with the loading of the rating vehicle introduces inaccuracy into the determination of ultimate shear capacity which in turn adversely affects the load rating of section shear. This section presents the effect of using the shear by proportionally increasing its value [5] for determining the ultimate shear capacity of a section. Figure 14 shows shear plotted against longitudinal strain. The ultimate strength lines are represented by $V_u$ and $\phi V_u$, where $\phi$ is the shear capacity reduction factor, for a potential critical shear section. Two scenarios of the shear value on the $V^*$ line are considered, one indicated by point A and the other by point B, the former with $V^*$ below and the latter with $V^*$ above the $V_u$ line. Let us assume that only moment and section shear are the influencing load effects. At point B, when the load effects $V^*$ and $M^*$ are known, the reduced ultimate shear capacity $\phi V_u$ shown using point B1 can be calculated without iteration. Caprani and Melhem [5] suggested an iteration process to determine the shear capacity by assuming proportional shear and the ratios of load effects (e.g., $M^*/V^*$, $N^*/V^*$ and $T^*/V^*$) remain constant. The solution-search procedure gives $V_u$(iteration) at point F, where the line $V^*$ intersects with the line $V_u$, with a corresponding $\phi V_u$ indicated by point F1. It can be seen that point F1, resulting in an unconservative calculation of $\phi V_u$ when compared with point B1. Using the same logic, for the scenario at point A, the iteration with proportionally increasing shear $V^*$ results in the same shear capacity $V_u$(iteration) at F and the $\phi V_u$(iteration) at F1, which is conservative compared to point A1.

For those scenarios where the load effects of the moving vehicle with $V^*$ values which fall between points D and E, where they fall below the $\phi V_u$ line suggesting adequate shear capacity. The assumed proportional loading for these $V^*$ values gives $\phi V_u$(iteration) indicated by point F1 after iterations. Since these $V^*$ values are greater than $\phi V_u$(iteration), load rating for these scenarios using the ultimate capacity determined for the proportional approach will show that these vehicle-bridge systems are inadequate for the assumed loading even though they are adequate for the actual loading of the moving vehicle as shown in the present approach.

Figure 15 shows the effect of using the proportionally increased shear to determine $V_u$ for the two span bridge beam with the rating vehicle described in Section 3.2. The trend of the longitudinal strain in this figure, caused by the moving vehicle, follows that of the magnitude of the bending moment as can be seen when this diagram is viewed together with Figure 9. This is because the influence of the bending moment on the strain is greater than that of the coexisting shear. At maximum $V^*$, the proportion of the total longitudinal strain from bending moment is 0.69.

Point A shown in the figure is the $V^*$ of 321 kN at maximum shear and point B is the corresponding $V_u$ (AS 5100.7) of 163 kN. The proportionally increased shear converges to a solution of 206 kN which is the intersection of line OA and the capacity line, indicated by point C. This figure shows that the $V_u$(iteration) at point C is not that of the rating vehicle moving on the bridge but for the vehicle at $x_{veh}$ = 16.5 m (the condition of point A on the diagram) with total factored load (consisting of both self weight of the beam and the

vehicular load) scaled by a ratio of $V_u$(iteration) at C to $V^*$(AS 5100.7) at A. This ratio is 206 kN/321 kN which gives a value of 0.64. Since at convergence, $V^*$(iteration) does not equal $V^*$(AS 5100.7), the ultimate shear capacity of 206 kN is for a vehicle-bridge system with the vehicle is at $x_{veh}$ = 16.5 m with either the safety coefficients or the nominal loads for both bridge and vehicle reduced by this proportion. This capacity 206 kN is greater than the actual capacity of 163 kN so the capacity used for rating the load effects at A is unconservative by 43%.

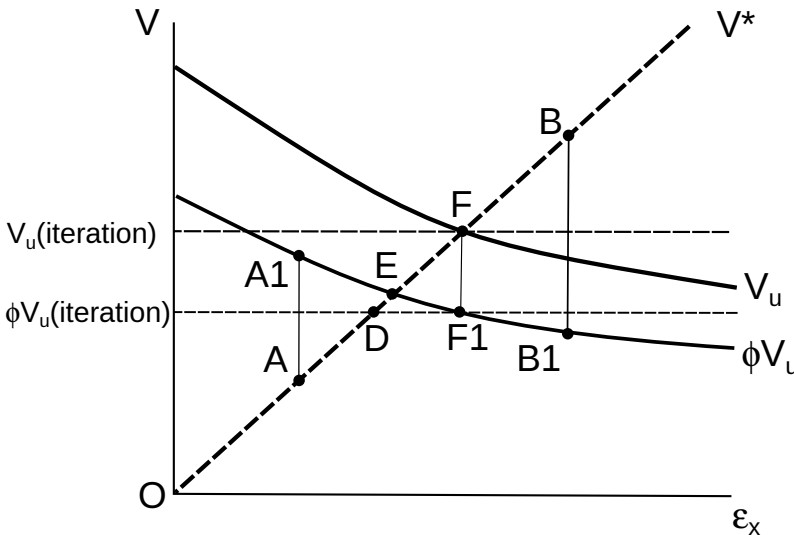

**Figure 14.** Adverse effects of finding ultimate shear using proportionally increased total load shear.

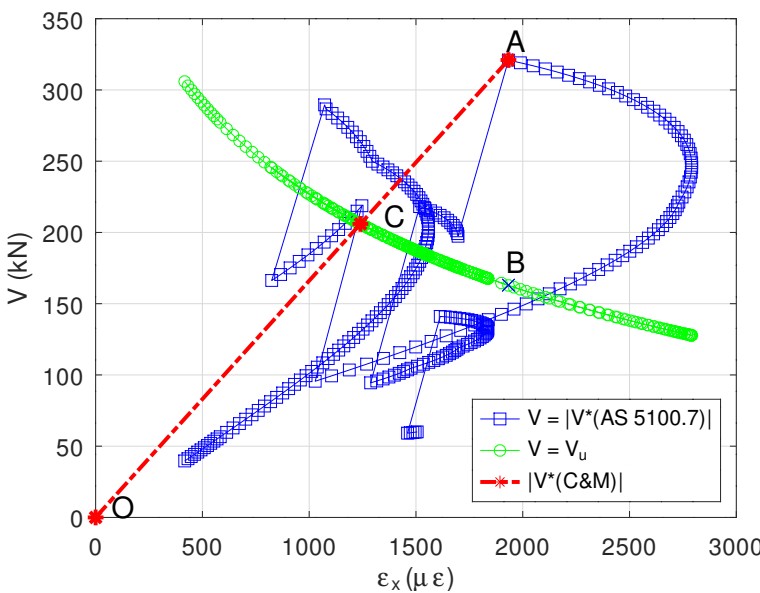

**Figure 15.** Effect of the proportionally increased shear on the two-span bridge beam.

The approach by Holt et al. [13] also introduced inaccuracy into the load rating. The reasons are similar to those described above. The difference in this approach from that of Caprani and Melhem's is that the proportional shear for live load, instead of total load, starts from point G (see Figure 16), instead of from the origin, and increases proportionally during the iteration process.

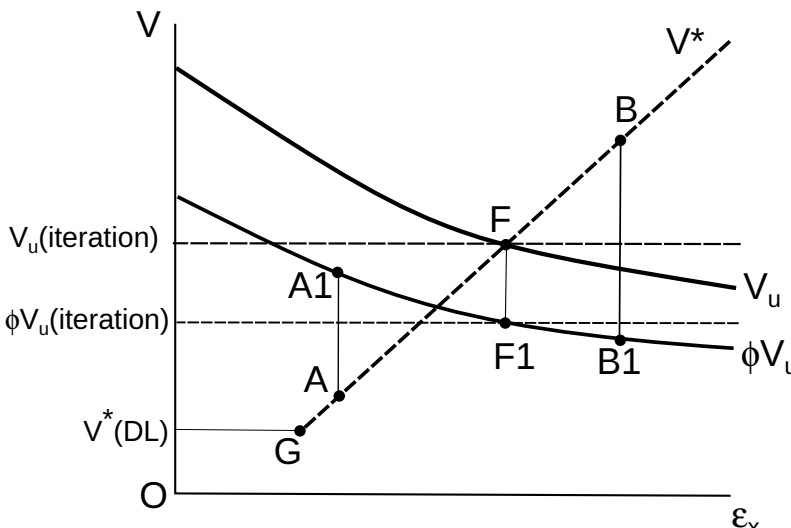

**Figure 16.** Adverse effects of finding ultimate shear using proportionally increased live load shear.

## 7. Conclusions

In the shear load rating for bridge assessment using MCFT-based provisions in a design standard, this paper proposes to determine the ultimate shear capacity directly from the load effects of the rating vehicle as stipulated in the standard. Two bridge beams, one simply supported and the other continuous, have been analysed to show the general characteristic of the load effects at a section close to a support. The results show that the variation of shear and associated load effects is quite complicated and cannot be modelled using the assumption of proportional factors in the load effects as proposed by several recent approaches. It has been demonstrated that using an assumed proportional loading to determine the ultimate shear capacity would result in inconsistency in the load effects to that of a moving vehicle, and thus an inaccuracy in the load rating.

This paper also identified shortcomings in AS 5100.7 and other national standards that use MCFT-based provisions for shear where the axial force in the longitudinal steel caused by several section load effects was not separately assessed for load rating. A new approach has been proposed to overcome these shortcomings, and to satisfy a key aspect of load rating, which is to load rate individual strength checks as stipulated in Clause 14.1 of AS 5100.7. This approach enables bridge authorities to specify appropriate strengthening requirements if needed.

Due to the complication in assessing the load effects and in determining the governing rating factor at every section of the bridge beam, the paper highlights the importance of automating the load rating by integrating it into the structural analysis to determine load effects.

**Author Contributions:** Conceptualization, K.W.W. and V.V.; methodology, K.W.W. and V.V.; software, K.W.W.; validation, K.W.W. and V.V.; formal analysis, K.W.W.; investigation, K.W.W. and V.V.; writing—original draft preparation, K.W.W.; writing—review and editing, K.W.W. and V.V. All authors have read and agreed to the published version of the manuscript.

**Funding:** This research received no external funding.

**Data Availability Statement:** Not applicable.

**Conflicts of Interest:** The authors declare no conflict of interest.

## Appendix A. Determination of Load Rating Factors

Equation (1) for $T_{td}$ is a function of $\Delta F_{td}$ which has non-linear, and therefore the longitudinal tension force due to live load, $T_{td}(\text{LL})$, cannot be calculated directly. It has to be calculated by subtracting the portion of $T_{td}$ due to dead load, $T_{td}(\text{DL})$, from $T_{td}(\text{DL\&LL})$,

as shown in the following calculation. Note that the values of $M^*$ and $V^*$ used in the calculations are absolute values. The terms DL, LL and DL&LL in brackets are to highlight that the values are obtained from dead load, live load, and dead load and live load respectively.

When $x_{veh}$ = 0.00 m, $M^*$(DL) = 9.90 kNm and $V^*$(DL) = 23.44 kN. From the analysis, $\phi V_{us}$(DL) = 109.22 kN and $\theta_v$(DL) = 29.63°.

When $x_{veh}$ = 20.81 m, $M^*$(DL&LL) = 153.49 kNm and $V^*$(DL&LL) = 382.42 kN, $\phi V_{us}$(DL&LL) = 76.50 kN and $\theta_v$(DL&LL) = 39.08°.

The lever arm (distance) between the forces $C$ and $T$ of the section in bending, $z$ = 0.39 m from the analysis.

$$
\begin{aligned}
\Delta F_{td}(\text{DL\&LL}) &= [V^*(\text{DL\&LL}) - 0.5 \times \phi V_{us}(\text{DL\&LL})] \times cot(\theta_v(\text{DL\&LL})) \\
&= [382.42 \text{ kN} - 0.5 \times 76.50 \text{ kN}] \times cot(39.08^o) \\
&= [382.42 \text{ kN} - 38.25 \text{ kN}] \times 1.231 \\
&= 423.67 \text{ kN (has to be } \geq 0) \\
&= 423.67 \text{ kN}
\end{aligned} \tag{A1}
$$

$$
\begin{aligned}
T_{td}(\text{DL\&LL}) &= \frac{M^*(\text{DL\&LL})}{z} + \Delta F_{td}(\text{DL\&LL}) \\
&= \frac{153.49 \text{ kNm}}{0.39 \text{ m}} + 423.67 \text{ kN} \\
&= 393.56 \text{ kN} + 423.67 \text{ kN} \\
&= 817.23 \text{ kN}
\end{aligned} \tag{A2}
$$

$$
\begin{aligned}
\Delta F_{td}(\text{DL}) &= [V^*(\text{DL}) - 0.5 \times \phi V_{us}(\text{DL})] \times cot(\theta_v(\text{DL})) \\
&= [23.44 \text{ kN} - 0.5 \times 109.22 \text{ kN}] \times cot(29.63^o) \\
&= [23.44 \text{ kN} - 54.61 \text{ kN}] \times 1.758 \\
&= -54.80 \text{ kN (has to be } \geq 0) \\
&= 0.0
\end{aligned} \tag{A3}
$$

$$
\begin{aligned}
T_{td}(\text{DL}) &= \frac{M^*(\text{DL})}{z} + F_{td}(\text{DL}) \\
&= \frac{9.90 \text{ kNm}}{0.39 \text{ m}} + 0.0 \\
&= 25.39 \text{ kN.}
\end{aligned} \tag{A4}
$$

$$
\begin{aligned}
T_{td.capacity} &= \phi \times f_{sy} \times A_{st} \\
&= 0.7 \times 500 \text{ N/mm}^2 \times 1350 \text{ mm}^2 \times 1 \times 10^{-3} \text{ kN/N} \\
&= 472.50 \text{ kN}
\end{aligned} \tag{A5}
$$

$$
\begin{aligned}
T_{td}(\text{LL}) &= T_{td}(\text{DL\&LL}) - T_{td}(\text{DL}) \\
&= 817.23 \text{ kN} - 25.39 \text{ kN} \\
&= 791.84 \text{ kN}
\end{aligned} \tag{A6}
$$

The load rating factor for force in the longitudinal steel in the flexural tension side of the section,

$$
\begin{aligned}
RF(\text{force}) &= \frac{T_{td.capacity} - T_{td}(\text{DL})}{T_{td}(\text{LL})} \\
&= \frac{472.50 \text{ kN} - 25.39 \text{ kN}}{791.84 \text{ kN}} \\
&= \frac{447.11 \text{ kN}}{791.84 \text{ kN}} \\
&= 0.57.
\end{aligned} \tag{A7}
$$

For completeness, the load rating factor for shear is calculated below.

$$V^*(\text{LL}) = V^*(\text{DL\&LL}) - V^*(\text{DL})$$
$$= 382.42 \text{ kN} - 23.44 \text{ kN} \tag{A8}$$
$$= 358.98 \text{ kN}$$

Thus, the load rating factor for shear,

$$RF(\text{shear}) = \frac{\phi V_u(\text{DL\&LL}) - V^*(\text{DL})}{V^*(\text{LL})}$$
$$= \frac{141.79 \text{ kN} - 23.44 \text{ kN}}{358.98 \text{ kN}} \tag{A9}$$
$$= \frac{118.35 \text{ kN}}{358.98 \text{ kN}}$$
$$= 0.33.$$

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
