# Peer review of "Determination of Shear Capacity for Load Rating of Concrete Bridges to AS 5100.7-2017"

_infrastructures, doi:10.3390/infrastructures7110156_

Round 1

Reviewer 1 Report

This paper describes several adverse effects of using the proportional load to determine the shear capacity for load rating. Numerical examples of two bridge beams, one simply supported and the other continuous, are presented to demonstrate that the characteristic of the load effects caused by a moving vehicle is not representable by proportional load effects. The present work is useful for the analysis of the ultimate shear capacity of a reinforced concrete. However, before the recommendation of this paper, the following comments should be considered.

1.      The wording formats should be clearly checked, some obvious mistakes occur in the first line of Abstract, line 87, etc. The format of the second paragraph in Introduction is inconsistent with others.

2.      Please ensure that you use boldface for matrices, vectors, and tensors; italics for all variables and lowercase Greek letters; and roman for all numerals, uppercase Greek characters, and mathematical operators. The unit symbol such as kN, m, mm, etc., should be normal rather than italics.

3.      Literature review is insufficient, at least some recent development has been reported on Load Effects from a moving Vehicle, which should be reviewed by including the reference “Recent Advances in Researches on Vehicle Scanning Method for Bridges

4.      The original drawing is suggested for the figures in the paper so as to improve the image pixel.

5.      The 3D plot in Figure 6 is unclear, which may be improved by rotating with a proper angle.

6.      Line 338, please use the correct expression on Equations, what is meaning is Equation 8.2.8.2(1)?

7.      The data listed on Page 11 are disorder and unreadable, which is suggested to be included in a table.

8.      Please give a clear definition for terms Vu and φVu.

9. More concise and useful conclusion should be given.

Author Response

The authors would like to thank the reviewer for all the comments.  The response to the review report is attached.

Reviewer 2 Report

The shear capacity of conrete bridges is examined and discussed accounting for the loading rate in the manuscript. The paper is well organized. The content is interesting and important to the design of bridge structures. Some minor modifications are required to further imporve the quality of the paper.

(1) The quality of figures is not very good, such as Figures 1, 4, 5, 6 etc.. It is suggested to improve the quality of these figures.

(2) The conclusions should be used to describe the contributions or new findings from the study. It is suggested to give these new findings point by point.

Author Response

(The authors gave the same response as above.)

Reviewer 3 Report

The author presented a new approach to the load rating that separately accounts for the load effect for axial failure mode of the longitudinal steel.What's more, it is pointed out that locating the critical section where the rating factor is minimum  can be automated by integrating load rating into the analysis of load effects. The results were well exposed and well represented as well as their analysis and interpretation except that more details are lacking on the approach adopted by the authors.

Therefore, I will ask the authors for some observations and questions:

1. Line 22 and line127, the abbreviation MCFT is stated in the abstract, please check it. And check the normality of other abbreviations.

2. Line 199-218, please add references.

3. Line247-248, the text formatting in Figure 3 should be consistent with Figure 1 and Figure 2.

4. Line345-346, please check the format of Equation 3 to make it clearer for readers.

5. Line 492-496, please analyze the applicability of these methods.

6. In section 3.2, please add the reinforcement arrangement diagram of the concrete beam.

Author Response

(The authors gave the same response as above.)

Round 2

Reviewer 1 Report

No more comments.

Author Response

Thank you for the Review Report Round 2.